# Real-world patterns in remote longitudinal study participation: A study of the Swiss Multiple Sclerosis Registry

**Paola Daniore**[1,2,3], **Chuqiao Yan**[4], **Mina Stanikic**[1,5], **Stefania Iaquinto**[5], **Sabin Ammann**[5], **Christian P. Kamm**[6,7], **Chiara Zecca**[8,9], **Pasquale Calabrese**[10], **Nina Steinemann**[5], **Viktor von Wyl**[1,2,5]*

1 Institute for Implementation Science in Health Care, University of Zurich, Zurich, Switzerland, 2 Digital Society Initiative, University of Zurich, Zurich, Switzerland, 3 Center for Digital Trust, Swiss Federal Institute of Technology Lausanne, Lausanne, Switzerland, 4 Institute of Applied Information Technology, Zurich University of Applied Sciences, Winterthur, Switzerland, 5 Epidemiology, Biostatistics and Prevention Institute, University of Zurich, Zurich, Switzerland, 6 Department of Neurology, Inselspital, Bern University Hospital, University of Bern, Switzerland, 7 Neurocentre, Luzerner Kantonsspital, Lucerne, Switzerland, 8 Faculty of Biomedical Sciences, Università della Svizzera Italiana (USI), Lugano, Switzerland, 9 Department of Neurology, Multiple Sclerosis Center, Neurocenter of Southern Switzerland, EOC, Lugano, Switzerland, 10 Neuropsychology and Behavioral Neurology Unit, Division of Cognitive and Molecular Neuroscience, University of Basel, Switzerland

* viktor.vonwyl@uzh.ch

**Data Availability Statement:** The underlying data for this analysis are human research participant

## Abstract

Remote longitudinal studies are on the rise and promise to increase reach and reduce participation barriers in chronic disease research. However, maintaining long-term retention in these studies remains challenging. Early identification of participants with different patterns of long-term retention offers the opportunity for tailored survey adaptations. Using data from the online arm of the Swiss Multiple Sclerosis Registry (SMSR), we assessed sociodemographic, health-related, and daily-life related baseline variables against measures of long-term retention in the follow-up surveys through multivariable logistic regressions and unsupervised clustering analyses. We further explored follow-up survey completion measures against survey requirements to inform future survey designs. Our analysis included data from 1,757 participants who completed a median of 4 (IQR 2–8) follow-up surveys after baseline with a maximum of 13 possible surveys. Survey start year, age, citizenship, MS type, symptom burden and independent driving were significant predictors of long-term retention at baseline. Three clusters of participants emerged, with no differences in long-term retention outcomes revealed across the clusters. Exploratory assessments of follow-up surveys suggest possible trends in increased survey complexity with lower rates of survey completion. Our findings offer insights into characteristics associated with long-term retention in remote longitudinal studies, yet they also highlight the possible influence of various unexplored factors on retention outcomes. Future studies should incorporate additional objective measures that reflect participants' individual contexts to understand their ability to remain engaged long-term and inform survey adaptations accordingly.

data and in combination potentially identifiable. The data that supports the findings of this study are available upon reasonable request. Requests should be directed to the Swiss Multiple Sclerosis Registry at the University of Zurich; Epidemiology, Biostatistics & Prevention Institute; Hirschengraben 84, CH-8001 Zurich (ms-register@ebpi.uzh.ch).

**Funding:** The author(s) received no specific funding for this work.

**Competing interests:** CPK has received honoraria for lectures as well as research support from Biogen, Novartis, Almirall, Teva, Merck, Sanofi Genzyme, Roche, Janssen, Eli Lilly, Celgene and the Swiss MS Society (SMSG). MS reports employment by Roche branch in Serbia, Roche d. o.o., from February 2019 to February 2020. CZ receives financial support for speaking, educational, research or travel grants from Abbvie, Almirall, Biogen Idec, Celgene, Sanofi, Merck, Novartis, Teva Pharma, Roche.

# Introduction

Multiple sclerosis (MS) is a chronic neurodegenerative autoimmune condition characterized by a wide range of symptom patterns and disease courses that remain incompletely understood [1]. Longitudinal epidemiological studies can offer a more comprehensive understanding of the associations between risk factors and certain symptom patterns or disease courses in people living with MS (pwMS), enabling more tailored treatment approaches [2]. Remote longitudinal studies, or longitudinal epidemiological studies delivered entirely online, hold promise for further advancing MS research by increasing reach and reducing participation barriers for pwMS related to time, accessibility and in-person presence [3–5]. In turn, studies can benefit from higher external validity, which enhances their real-world application.[6]

Remote longitudinal studies have gained considerable traction in chronic disease research due to their efficient and innovative approaches to engage participants [7,8]. Nevertheless, such studies are still prone to low long-term retention, which increases the risk of selection bias in the case of a systematic difference between participants lost to follow-up and those left in the study [9,10]. The resulting methodological challenges in such studies [11,12] call for the identification of participant subgroups associated with varying levels of long-term retention in order to inform novel survey designs [13]. In MS research, recent digital health intervention studies suggest that subgroups defined by sociodemographic characteristics, including age and gender, increased symptom burden from disease progression, and daily life responsibilities such as employment, can influence study participation [2,3,14,15].

However, there is currently limited experience in identifying subgroups of pwMS with varying retention levels in remote longitudinal studies. This highlights the need to explore new methods for identifying subgroups that can help ensure long-term retention in such studies [16]. In turn, these findings can guide the design of future online surveys for MS research by allowing for tailored study designs specific to different subgroups.

## Aims

We aimed to apply well-established statistical methods in health research to explore (1) baseline participant characteristics associated with long-term retention in the online arm of the Swiss Multiple Sclerosis Registry (SMSR) and (2) the influence of survey design elements on participants' survey completion. Through these, we aimed to demonstrate different analyses that can be performed at baseline to identify population subgroups with distinct response patterns and inform survey adaptations accordingly. The overarching goal of this study was to inform future remote longitudinal study design to facilitate long-term retention in MS as well as other chronic disease research.

## Methods

### Setting

This study uses data from participants registered in the SMSR, an observational study initiated in 2016, with ongoing enrollment. The SMSR recruits adult pwMS residing or receiving treatment in Switzerland. Participation in the SMSR is voluntary and requires completion of a signed consent form along with a confirmation of MS or clinically isolated syndrome (CIS) diagnosis. Upon enrollment, all participants complete a baseline questionnaire and continue their participation by providing self-reported data through follow-up questionnaires provided approximately twice a year, accessible both online and in paper format. All participants who

fill out a baseline questionnaire receive an invitation to fill out follow-up and thematic surveys. The SMSR was approved by the Ethics Committee of the Canton of Zurich (PB-2016–00,894; BASEC-NR 2019–01,027). Additional information about the SMSR can be found in prior publications [17,18].

This study refers to data from the online format of the baseline survey, along with all online follow-up and thematic surveys from the SMSR, hereafter referred to as *follow-up surveys*. Participants receive regular invitations and reminders to fill out the questionnaires, and are given the option to pause and recontinue the completion of the surveys on their own schedules. The first follow-up survey (*FU-6*) was released in June 2017, and the last follow-up survey considered in this analysis (*FU-48*) was released in October 2022. We refer to data from June 2016 to May 2023. Participants with complete baseline sociodemographic information, including age, sex, living situation, education, citizenship, and partnership are included in the study analysis (S1 Fig).

## Outcome variables

We developed two primary outcome variables from responses to online *follow-up surveys*. The variables were informed by previous studies [19–21] and were used to create a binary classification of online participation in the SMSR. The first primary outcome variable categorized participants under *high* or *low retention* based on whether they completed at least one follow-up survey each calendar year, after participating in the SMSR for two years. We selected a two-year minimum for participation to ensure an adequate observational period for individuals with chronic conditions [22]. The second primary outcome variable categorized participants under *lower* or *higher response than median* based on whether the number of completed surveys was equal to or above, or below the median number of surveys completed by participants who joined the SMSR in the same year. Sensitivity analyses were done using a variation of the first outcome variable. This variation was based on whether the participants completed at least one follow-up survey each calendar year until the end of the study period (2022 or 2023), to assess trends in active participation from the start to the end of the survey period.

## Variables of interest at baseline

This study evaluated three categories of baseline data identified in previous studies on MS and chronic diseases [23–28] that can contribute to different patterns in retention, namely *sociodemographic*, *health-related* and *daily-life* variables. The sociodemographic variables include age, sex, Swiss language regions, whether participants have children, whether they received a university degree, partnership status, and whether they have a Swiss citizenship. The health-related variables include variables that define the health status and indicators of disease development at baseline. These include MS type, years since MS diagnosis, whether participants have relatives with MS, symptoms of fatigue, symptoms of paraesthesia, symptoms of depression and self-reported disability status scale [29] (SRDSS). The daily-life variables provide contextual indicators for partaking in daily activities. These include whether participants independently drive a car, use public transport, employment status and whether they received assistance to fill out the surveys. Sensitivity analyses were performed to assess whether the timing of participants receiving disease-modifying therapies, or not receiving them at all, influenced the main analyses.

## Statistical analyses

In this study, we describe participant characteristics and primary outcomes of interest using counts and percentages, as well as median and interquartile range (IQR). To identify predictors

for long-term retention at baseline, univariable and multivariable logistic regression analyses were conducted for both primary outcomes. All a priori selected baseline variables were included in these analyses. Additional interactions were explored and kept if they decreased the Akaike information criterion (AIC) score by 2 points or more [30–32]. Events per variable (EPV) were calculated to ensure a minimum of 10 EPV for sample size considerations in logistic regression analyses [33]. Multicollinearity was evaluated using the Variance Inflation Factor (VIF), with VIF values above 5 indicating collinearity [34]. Finally, variables with perfect separation were recategorized into fewer categories to balance the responses [35]. Since 155 participants had partially missing data, complete case analyses as well as multiple imputed data derived from 5 multiple imputed datasets were analysed for comparison (S1 Fig) [36].

We conducted an exploratory k-means clustering analysis to identify possible participant subgroups at baseline with varying long-term retention. The clustering analysis was conducted in two steps: (1) unsupervised clustering using all baseline variables for participants with complete data used in the primary analysis, and (2) associating the identified clusters of participants with their respective long-term retention outcomes. For the clustering analysis, categorical variables were converted into binary variables for each category level, and continuous variables were standardized to minimize the impact of scale differences on model fitting. The elbow method was used to determine the optimal number of clusters (S2 Fig). We finally conducted exploratory descriptive analyses on survey design factors, including question length and Flesch-Kinkaid scores [37] for text readability, and their associations with follow-up survey completion. Survey completion is defined by responses to the first and last survey questions, as well as responses to an equal or greater number of questions compared to the median number of responses from all survey participants.

All analyses were conducted in R (version 4.2.2) and Python (version 3.12.0).

## Results

### Study population and response

A total of 1,757 participants were included in the analyses (flowchart available in S1 Fig). Sociodemographic, health, and daily life related characteristics of the participants for both primary outcomes of interest are reported in Table 1. Overall, 671 (38%) participants filled out at least one survey per year during the first two years of study participation and 948 (54%) participants filled out an equal number or more surveys compared to those who completed the baseline questionnaire in the same starting year. During the study period, 41 (2.3%) left the SMSR by explicitly deregistering or dying (S1 Table). Median age [IQR] for participants was 45 [36–54] years. Most participants were female (n = 1,254, 71%), resided in the German/Romansh language region of Switzerland (n = 1,357, 77%), had relapsing-remitting MS (n = 1,257, 72%) and were employed at baseline (n = 1,173, 67%). Characteristics of participants for the sensitivity analysis with an alternate outcome variable are available in S2 Table.

Overall, participants completed a median of 4 (IQR 2–8) follow-up surveys after baseline, with a maximum of 13 surveys, which corresponds to the total number of follow-up surveys in the study period (Fig 1).

### Baseline predictors for long-term participation: yearly retention

Multivariable regression analyses for the complete case data (Fig 2 and S3 Table) reveal similar results to univariable regression analyses for the complete case data and multivariable regression analyses for multiple-imputed data (S3 Table). Compared to participants who joined the MS Registry in 2016, participants were less likely to actively participate on a yearly basis if they filled out the baseline questionnaire between 2017 and 2019 (OR 0.77, 95% CI [0.61, 0.97]) or

**Table 1. Baseline characteristics of participants for both primary outcomes of interest.**

| | Primary outcome 1: yearly retention | | Primary outcome 2: starting year-based retention | | Overall (N = 1757) |
|---|---|---|---|---|---|
| | Low retention (N = 1086) | High retention (N = 671) | Lower than median (N = 809) | Higher than median (N = 948) | |
| **Age** | | | | | |
| 18–35 | 294 (27.1%) | 134 (20.0%) | 227 (28.1%) | 201 (21.2%) | 428 (24.4%) |
| 36–45 | 288 (26.5%) | 176 (26.2%) | 198 (24.5%) | 266 (28.1%) | 464 (26.4%) |
| 46–55 | 296 (27.3%) | 211 (31.4%) | 218 (26.9%) | 289 (30.5%) | 507 (28.9%) |
| 56–65 | 149 (13.7%) | 111 (16.5%) | 115 (14.2%) | 145 (15.3%) | 260 (14.8%) |
| 66 and older | 59 (5.4%) | 39 (5.8%) | 51 (6.3%) | 47 (5.0%) | 98 (5.6%) |
| **Sex** | | | | | |
| Male | 312 (28.7%) | 191 (28.5%) | 229 (28.3%) | 274 (28.9%) | 503 (28.6%) |
| Female | 774 (71.3%) | 480 (71.5%) | 580 (71.7%) | 674 (71.1%) | 1254 (71.4%) |
| **Language region** | | | | | |
| German / Romansch | 812 (74.8%) | 545 (81.2%) | 600 (74.2%) | 757 (79.9%) | 1357 (77.2%) |
| French | 195 (18.0%) | 96 (14.3%) | 142 (17.6%) | 149 (15.7%) | 291 (16.6%) |
| Italian | 39 (3.6%) | 19 (2.8%) | 31 (3.8%) | 27 (2.8%) | 58 (3.3%) |
| Missing | 40 (3.7%) | 11 (1.6%) | 36 (4.5%) | 15 (1.6%) | 51 (2.9%) |
| **Survey start year** | | | | | |
| 2016 | 290 (26.7%) | 244 (36.4%) | 267 (33.0%) | 267 (28.2%) | 534 (30.4%) |
| 2017–2019 | 573 (52.8%) | 350 (52.2%) | 428 (52.9%) | 495 (52.2%) | 923 (52.5%) |
| 2020 onwards | 223 (20.5%) | 77 (11.5%) | 114 (14.1%) | 186 (19.6%) | 300 (17.1%) |
| **Has children** | | | | | |
| Yes | 574 (52.9%) | 378 (56.3%) | 445 (55.0%) | 507 (53.5%) | 952 (54.2%) |
| No | 512 (47.1%) | 293 (43.7%) | 364 (45.0%) | 441 (46.5%) | 805 (45.8%) |
| **Has university degree** | | | | | |
| Yes | 337 (31.0%) | 219 (32.6%) | 234 (28.9%) | 322 (34.0%) | 556 (31.6%) |
| No | 749 (69.0%) | 452 (67.4%) | 575 (71.1%) | 626 (66.0%) | 1201 (68.4%) |
| **Civilian status** | | | | | |
| Partnership / married | 541 (49.8%) | 365 (54.4%) | 405 (50.1%) | 501 (52.8%) | 906 (51.6%) |
| Not in a partnership | 545 (50.2%) | 306 (45.6%) | 404 (49.9%) | 447 (47.2%) | 851 (48.4%) |
| **Living situation** | | | | | |
| Living with spouse / family / friends / community | 827 (76.2%) | 528 (78.7%) | 615 (76.0%) | 740 (78.1%) | 1355 (77.1%) |
| Living alone / Single-parenting | 259 (23.8%) | 143 (21.3%) | 194 (24.0%) | 208 (21.9%) | 402 (22.9%) |
| **Swiss citizenship** | | | | | |
| Yes | 933 (85.9%) | 615 (91.7%) | 705 (87.1%) | 843 (88.9%) | 1548 (88.1%) |
| No | 153 (14.1%) | 56 (8.3%) | 104 (12.9%) | 105 (11.1%) | 209 (11.9%) |
| **Years since MS diagnosis** | | | | | |
| Mean (SD) | 8.70 (8.87) | 9.25 (8.80) | 9.28 (9.16) | 8.60 (8.57) | 8.92 (8.85) |
| Median [Min, Max] | 6.00 [0, 49.0] | 7.00 [0, 48.0] | 6.00 [0, 49.0] | 6.00 [0, 48.0] | 6.00 [0, 49.0] |
| Missing | 30 (2.8%) | 15 (2.2%) | 19 (2.3%) | 26 (2.7%) | 45 (2.6%) |
| **MS Type** | | | | | |
| RRMS | 797 (73.4%) | 460 (68.6%) | 575 (71.1%) | 682 (71.9%) | 1257 (71.5%) |
| CIS | 30 (2.8%) | 16 (2.4%) | 20 (2.5%) | 26 (2.7%) | 46 (2.6%) |
| PPMS | 99 (9.1%) | 62 (9.2%) | 79 (9.8%) | 82 (8.6%) | 161 (9.2%) |
| SPMS / Transition | 139 (12.8%) | 123 (18.3%) | 119 (14.7%) | 143 (15.1%) | 262 (14.9%) |
| Missing | 21 (1.9%) | 10 (1.5%) | 16 (2.0%) | 15 (1.6%) | 31 (1.8%) |
| **MS in relatives** | | | | | |

*(Continued)*

**Table 1.** (Continued)

| | Primary outcome 1: yearly retention | | Primary outcome 2: starting year-based retention | | Overall (N = 1757) |
|---|---|---|---|---|---|
| | Low retention (N = 1086) | High retention (N = 671) | Lower than median (N = 809) | Higher than median (N = 948) | |
| Yes | 217 (20.0%) | 128 (19.1%) | 162 (20.0%) | 183 (19.3%) | 345 (19.6%) |
| No | 869 (80.0%) | 543 (80.9%) | 647 (80.0%) | 765 (80.7%) | 1412 (80.4%) |
| **Symptoms: fatigue** | | | | | |
| Yes | 659 (60.7%) | 386 (57.5%) | 506 (62.5%) | 539 (56.9%) | 1045 (59.5%) |
| No | 427 (39.3%) | 285 (42.5%) | 303 (37.5%) | 409 (43.1%) | 712 (40.5%) |
| **Symptoms: paresthesia** | | | | | |
| Yes | 528 (48.6%) | 351 (52.3%) | 384 (47.5%) | 495 (52.2%) | 879 (50.0%) |
| No | 558 (51.4%) | 320 (47.7%) | 425 (52.5%) | 453 (47.8%) | 878 (50.0%) |
| **Symptoms: depression** | | | | | |
| Yes | 154 (14.2%) | 70 (10.4%) | 113 (14.0%) | 111 (11.7%) | 224 (12.7%) |
| No | 932 (85.8%) | 601 (89.6%) | 696 (86.0%) | 837 (88.3%) | 1533 (87.3%) |
| **Symptom burden** | | | | | |
| 1–3 symptoms | 350 (32.2%) | 196 (29.2%) | 249 (30.8%) | 297 (31.3%) | 546 (31.1%) |
| 4–6 symptoms | 259 (23.8%) | 191 (28.5%) | 202 (25.0%) | 248 (26.2%) | 450 (25.6%) |
| More than 7 symptoms | 320 (29.5%) | 185 (27.6%) | 246 (30.4%) | 259 (27.3%) | 505 (28.7%) |
| No symptoms | 157 (14.5%) | 99 (14.8%) | 112 (13.8%) | 144 (15.2%) | 256 (14.6%) |
| **SRDSS score** | | | | | |
| SRDSS 0–3.5 | 818 (75.3%) | 477 (71.1%) | 582 (71.9%) | 713 (75.2%) | 1295 (73.7%) |
| SRDSS 4–6.5 | 180 (16.6%) | 135 (20.1%) | 148 (18.3%) | 167 (17.6%) | 315 (17.9%) |
| SRDSS > = 7 | 70 (6.4%) | 39 (5.8%) | 58 (7.2%) | 51 (5.4%) | 109 (6.2%) |
| Missing | 18 (1.7%) | 20 (3.0%) | 21 (2.6%) | 17 (1.8%) | 38 (2.2%) |
| **Receives disability insurance** | | | | | |
| Yes | 317 (29.2%) | 193 (28.8%) | 261 (32.3%) | 249 (26.3%) | 510 (29.0%) |
| No | 769 (70.8%) | 478 (71.2%) | 548 (67.7%) | 699 (73.7%) | 1247 (71.0%) |
| **Currently drives** | | | | | |
| Yes | 847 (78.0%) | 529 (78.8%) | 607 (75.0%) | 769 (81.1%) | 1376 (78.3%) |
| No | 239 (22.0%) | 142 (21.2%) | 202 (25.0%) | 179 (18.9%) | 381 (21.7%) |
| **Uses public transport** | | | | | |
| Yes | 974 (89.7%) | 603 (89.9%) | 710 (87.8%) | 867 (91.5%) | 1577 (89.8%) |
| No | 112 (10.3%) | 68 (10.1%) | 99 (12.2%) | 81 (8.5%) | 180 (10.2%) |
| **Currently working** | | | | | |
| Yes | 734 (67.6%) | 439 (65.4%) | 518 (64.0%) | 655 (69.1%) | 1173 (66.8%) |
| No | 352 (32.4%) | 232 (34.6%) | 291 (36.0%) | 293 (30.9%) | 584 (33.2%) |
| **Someone helped with survey** | | | | | |
| Yes | 53 (4.9%) | 36 (5.4%) | 53 (6.6%) | 36 (3.8%) | 89 (5.1%) |
| No | 1033 (95.1%) | 635 (94.6%) | 756 (93.4%) | 912 (96.2%) | 1668 (94.9%) |

Results are presented in number (n) and percentage (%) for categorical variables and mean (SD) or median (IQR) for continuous variables; RRMS, Relapsing remitting MS; CIS, Clinically Isolated Syndrome; PPMS, Primary progressive MS; SPMS, Secondary progressive MS; SRDSS, self-reported disability status scale.

from 2020 onwards (OR 0.40, 95% CI [0.28, 0.57]). For the sociodemographic characteristics, compared to participants aged between 18–35 years old at baseline, participants in the 36–45 year old age group (OR 1.60, 95% CI [1.16, 2.21]) and 46–55 year old age group (OR 1.58, 95% CI [1.13, 2.20]) were more likely to actively participate on a yearly basis, while Swiss citizens

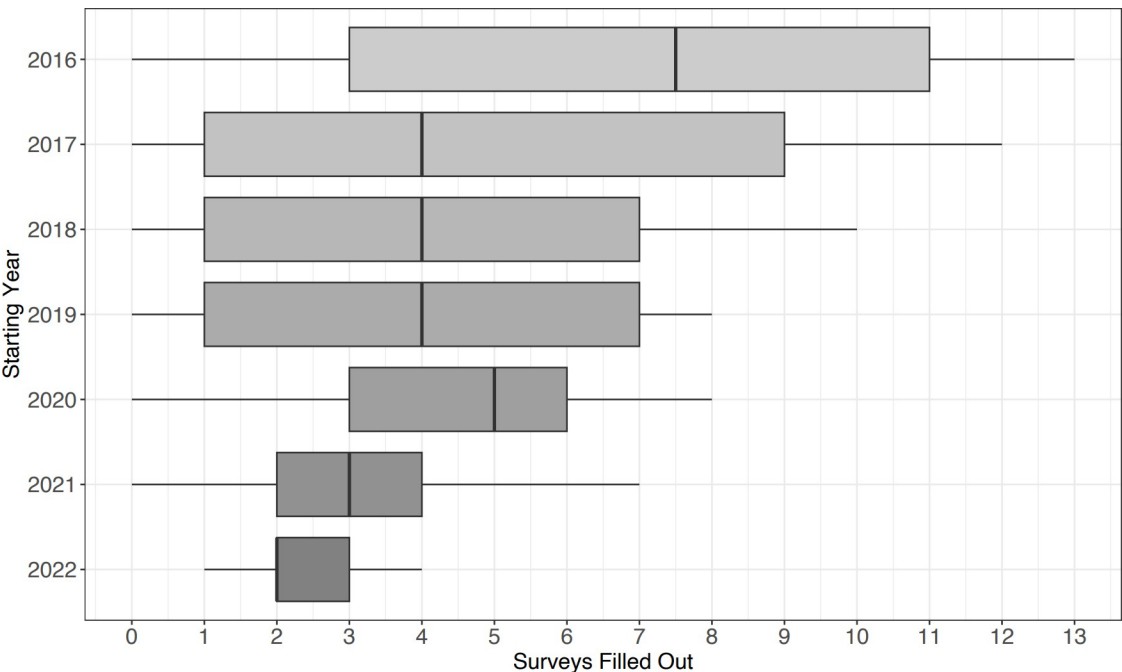

**Fig 1. Surveys filled out by starting year.**

revealed more likely to actively participate on a yearly basis that non-Swiss citizens (OR 1.60, 95% CI [1.11, 2.34]). For the health-related characteristics, compared to participants with relapsing remitting MS, participants with the more advanced secondary progressive MS or in transition were more likely to actively participate on a yearly basis (OR 1.57, 95% CI [1.08, 2.28]). Sensitivity analyses that included the timing variable for participants receiving disease-modifying therapies showed no differences in the results (**S4 Table**).

The three identified clusters of participants (**Figs 3** and **S3 and S5 Table**) also revealed sub-groups based on age groups, with one subgroup (cluster 1, n = 456) consisting primarily of participants aged 56+ (48%), another subgroup of participants (cluster 3, n = 608) between the ages of 36–55 (72%), and the last subgroup (cluster 2, n = 538) with participants between the ages of 18–35 (52%). Cluster 1 primarily comprises individuals with the secondary progressive or in transition form of MS (40%), while clusters 2 and 3 (89% and 86%, respectively) predominantly consist of individuals with the less advanced relapsing remitting MS. Counts of yearly high retention outcomes were 41% (95% CI [36%,45%]) for cluster 1, 37% (95% CI [33%,41%]) for cluster 2 and 39% (95% CI [35%,43%]) for cluster 3. As such, differences in long-term retention between the clusters were not observed (**S6 Table**).

### Baseline predictors for long-term participation: starting year-based retention

Multivariable regression analyses for the complete case data (**Fig 4** and **S7 Table**) revealed similar results to univariable regression analyses for the complete case data and multivariable regression analyses for multiple-imputed data (**S7 Table**). For the sociodemographic characteristics, compared to participants aged between 18–35 years old at baseline, participants in the 36–45 year old age group (OR 2.01, 95% CI [1.48, 2.74]), in the 46–55 year old age group (OR 2.11, 95% CI [1.53, 2.92]), in the 56–65 year old age group (OR 2.17, 95% CI [1.44, 3.30]) and in the 66 and over age group (OR 2.04, 95% CI [1.08, 3.87]) were more likely to participate

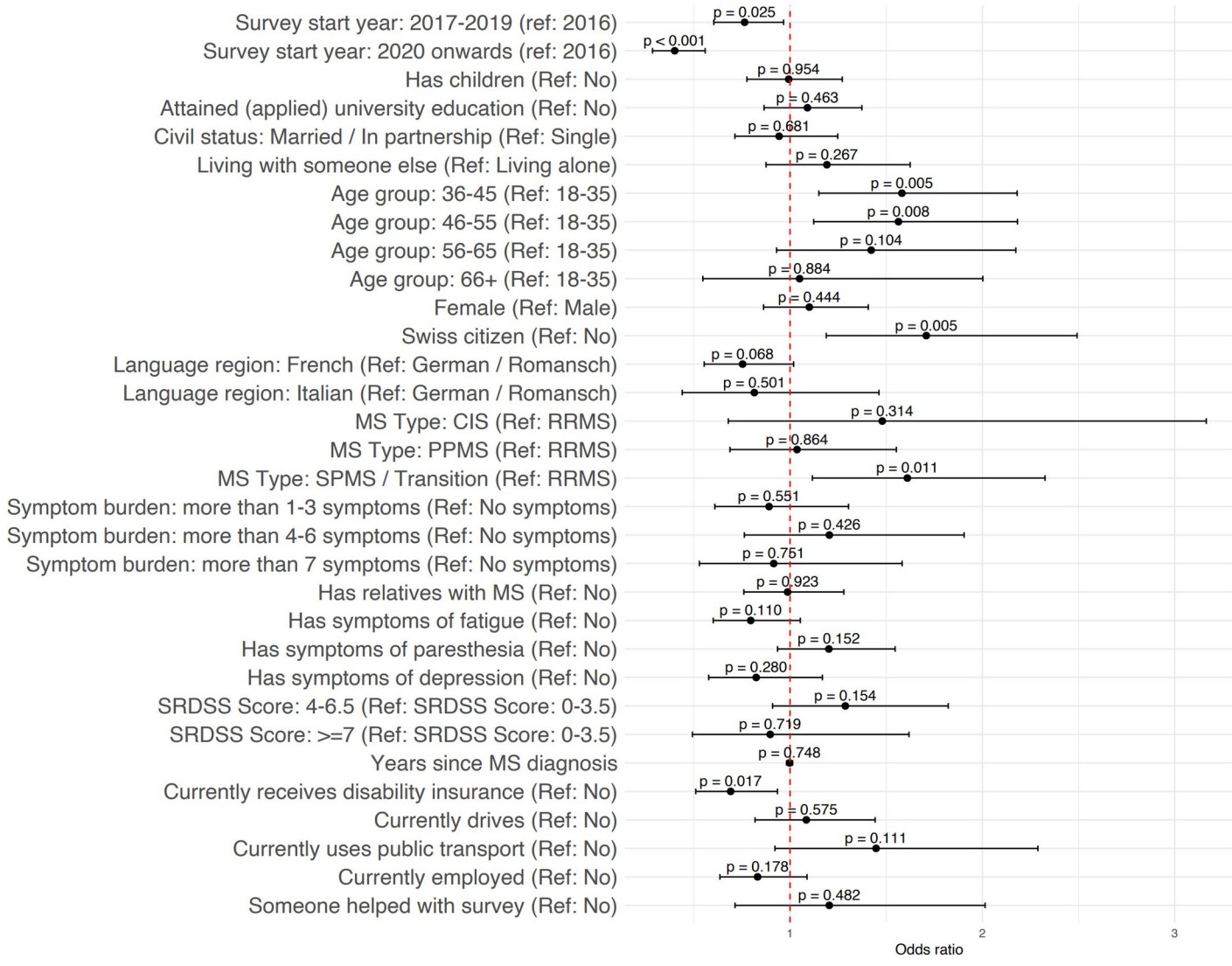

**Fig 2. Multivariable logistic regression analyses of predictors of yearly retention.**

than participants of the same starting year. For the health-related characteristics, participants with symptoms of fatigue were less likely to participate than those without symptoms of fatigue (OR 0.75, 95% CI [0.57, 0.99]). For the daily life-related characteristics, participants who regularly drove a car at baseline were more likely to participate than participants who did not drive a car (OR 1.38, 95% CI [1.04, 1.82]). Sensitivity analyses with an alternate outcome variable (S8 Table) further highlight the relevance of these characteristics on long-term retention. Sensitivity analyses that included the timing variable for participants receiving disease-modifying therapies showed no differences in the results (S9 Table).

The three identified clusters of participants (Figs 3 and S3 and S5 Table) also revealed subgroups based on age groups, with one subgroup consisting primarily of participants aged 56+ (cluster 1, 48%), another cluster of participants between the ages of 36–55 (cluster 3, 72%), and the last cluster with participants between the ages of 18–35 (cluster 2, 52%). Cluster 1 includes participants with a high burden of fatigue symptoms (79%) and a higher prevalence of individuals who do not drive (35%) or use public transport (28%) within the cluster compared to

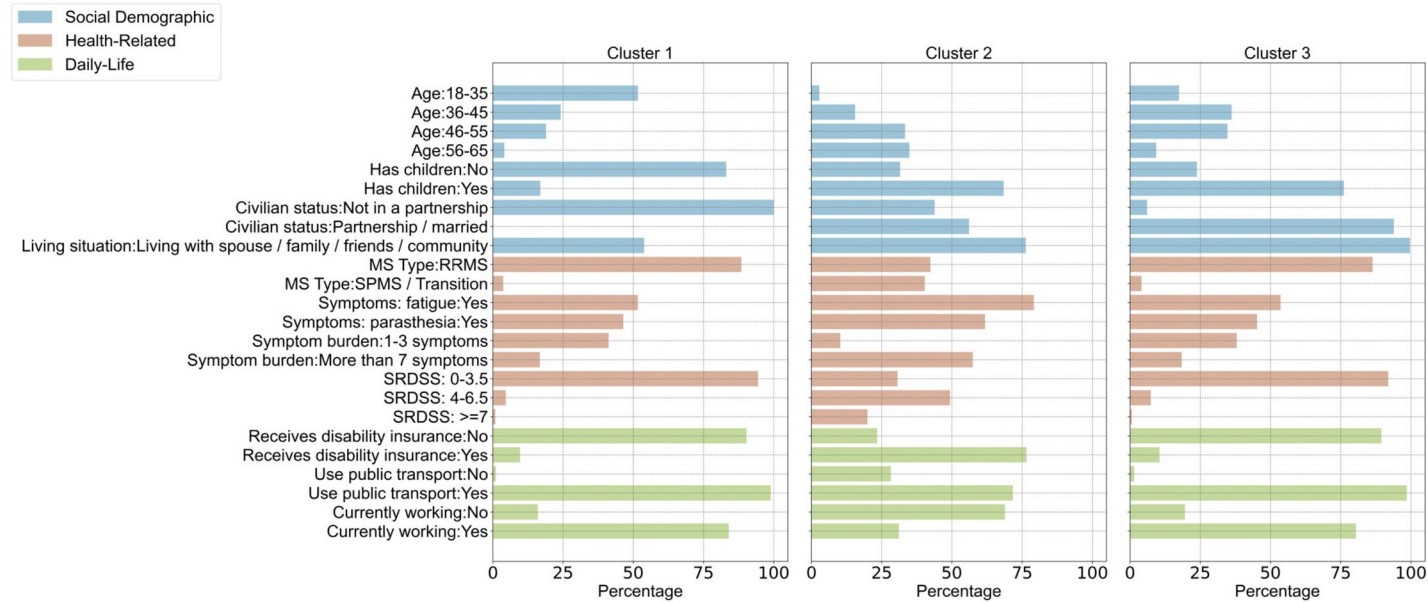

**Fig 3. Relative contribution of the variables in each cluster for the pre-defined characteristics (social demographic, health-related and daily life).** The bar chart illustrates the primary contributing variables to each cluster for the pre-defined characteristics represented as within-cluster percentages. Overall, cluster 1 is mostly comprised of participants in the older age groups, who have a more advanced form of MS, have higher symptom and disability burden, and are unemployed; cluster 2 is mostly comprised of participants from the youngest age groups, who are not in a partnership, do not have children, have a less advanced form of MS, have low symptom and disability burden and are unemployed; and cluster 3 is mostly comprised of participants in the middle age-group, who are in a partnership, have children and have similar health-related and daily-life characteristics as participants in cluster 2. These results are displayed as Sankey plots in **S3 Fig**.

clusters 2 and 3. Counts of high starting year-based retention outcomes were 52% (95% CI [48%, 57%]) for cluster 1, 55% (95% CI [51%, 59%]) for cluster 2 and 57% (95% CI [52%, 61%]) for cluster 3. As such, differences in long-term retention between the clusters were not observed (**S6 Table**).

## Complexity of follow-up surveys and impact on survey non-completion

Overall, the median length of the follow-up surveys was 29,468 characters (IQR: 17,712–36,673), 3,807 words (IQR: 2,271–4,918) and 70 questions (IQR: 59–78) (survey details and measures presented in **S10 Table**). Median readability of the surveys based on the Flesch-Kinkaid score was around the 7[th] grade school level (i.e., ages 12–13) (IQR: 6.2–7.6) (**S4 Fig**). The median percentage of participants who completed the surveys among those who started the surveys was 51% (IQR: 49–55%, **Fig 5**). A negative trend was observed between the completion of surveys for these participants and the total number of characters in a survey (correlation = -0.44), or the total number of questions in a survey (correlation = -0.33). However, both correlations were not statistically significant. The median percentage of participants who completed the surveys among those who were invited to fill out the surveys was 25% (IQR: 24–31%, **Fig 6**). A negative trend was also observed between the completion of surveys for these participants and the total number of characters in a survey (correlation = -0.36), or the total number of questions in a survey (correlation = -0.51). However, both correlations were also not statistically significant.

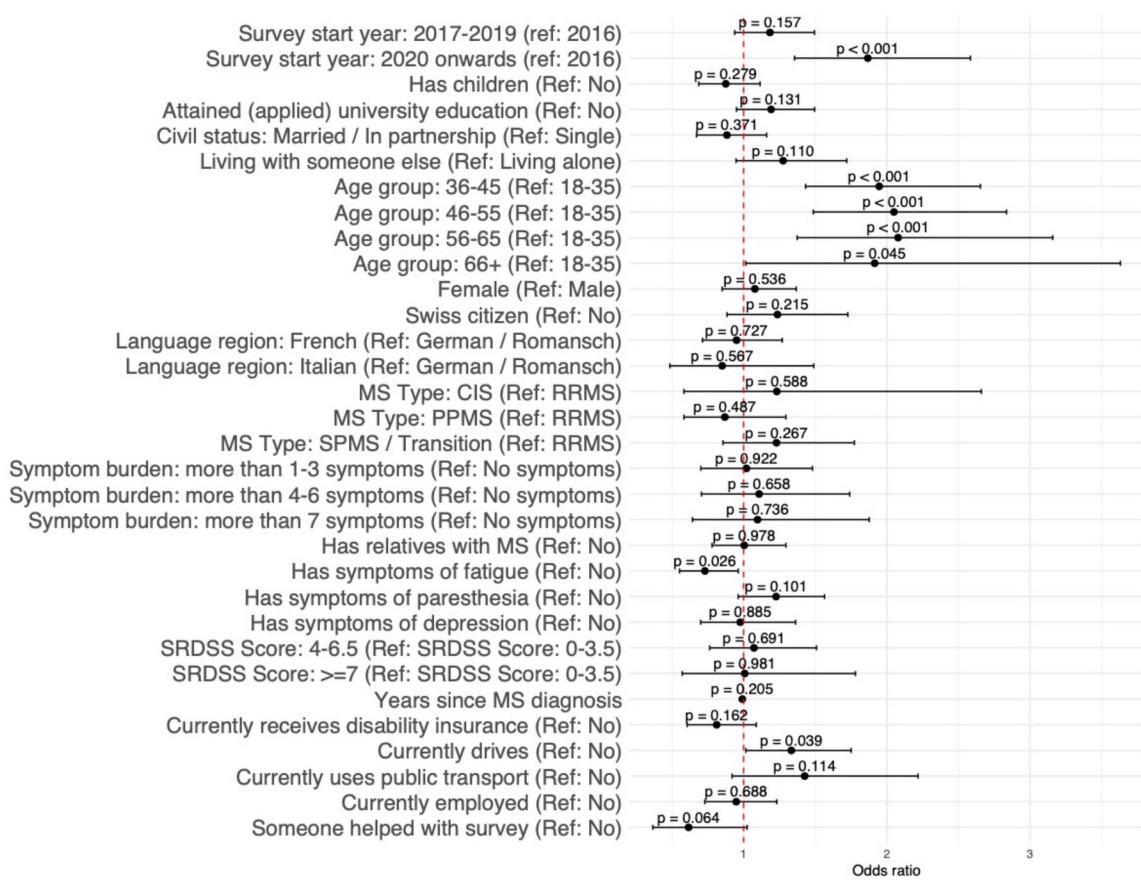

**Fig 4. Multivariable logistic regression analyses of predictors of higher long-term retention based on starting year.**

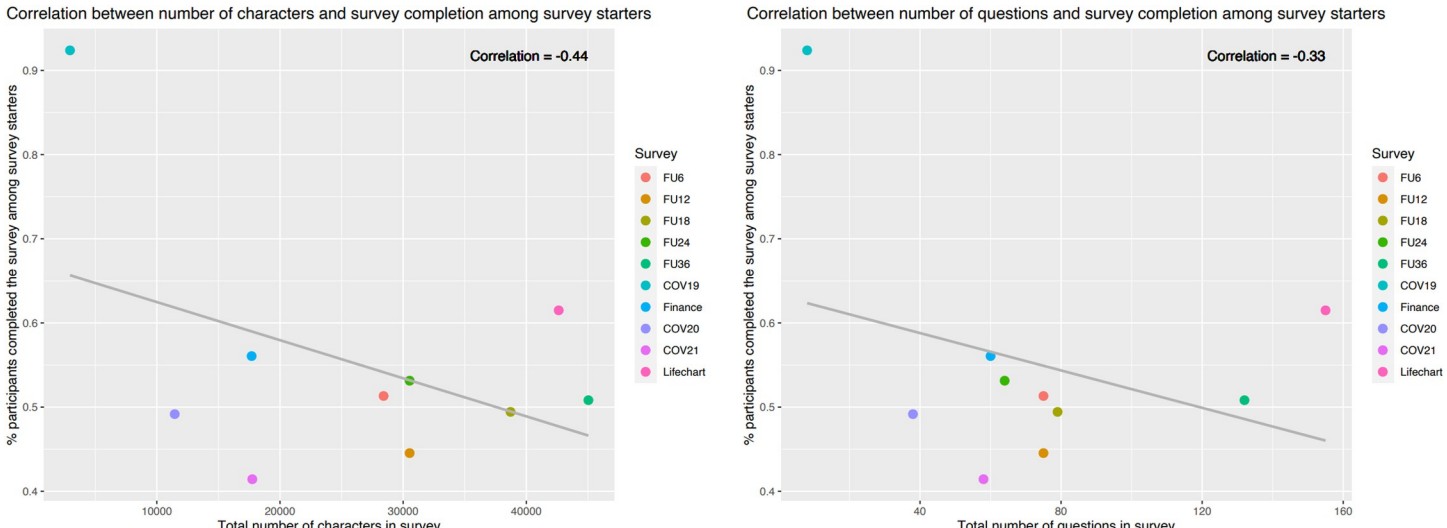

**Fig 5.** Survey complexity, displayed by number of characters (left) and number of questions (right) in the survey, vs percentage of participants who completed the survey among those who started the survey.

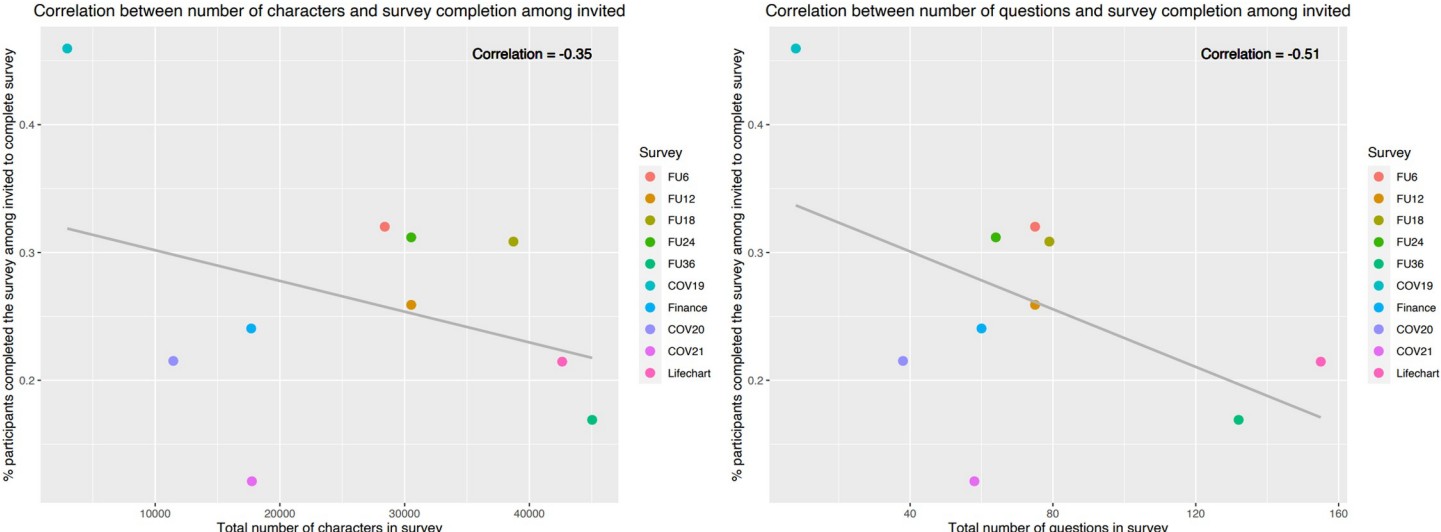

**Fig 6.** Survey complexity, displayed by number of characters (left) and number of questions (right) in the survey, vs percentage of participants who completed the survey among those who were invited to complete the survey.

## Discussion

This study reports findings on long-term participant retention using data from the online arm of the SMSR, a real-world longitudinal study on pwMS in Switzerland [17,18]. Our findings reveal that start year, age, citizenship, MS type, symptom burden and whether participants independently drive a car on a regular basis are associated with long-term retention at baseline. These findings were consistent for two different primary outcomes of interest reflective of long-term retention, with associations in the same direction of effect among variables within the three baseline categories for both analyses. Exploratory clustering of the baseline data revealed three distinct subgroups, however no differences were found in long-term retention among the clusters. Exploratory assessments of follow-up survey design characteristics and survey completion suggest the potential for adjusting survey components to enhance overall long-term survey participation.

Our analysis revealed that participants experiencing higher fatigue symptom burden and receiving disability insurance at baseline were less likely to be retained in the long-term. Participants with progressive MS and who drive a car independently showed a greater likelihood of long-term retention. These findings suggest that participants with progressive MS who retain independence may be more motivated to participate in research, likely due to reduced work commitments at this stage, as observed in a previous SMSR study.[38] Yet, these participants may be at risk of lower retention due to higher disability burden. Recent digital health interventions [2,39,40] and longitudinal surveys [41–43] assessing pwMS and other chronic diseases report similar lower retention for individuals with more pronounced symptom and disability severity. Notably, a recent analysis of a SMSR follow-up survey identified a subgroup of participants with low-frequency adoption of health technology with higher disability and disease burden, suggesting they may be more vulnerable to digital exclusion.[44] These individuals may be intrinsically motivated to join a study, likely due to the progression of MS, however could require adaptations to participate in the long-term [4,45,46]. Some approaches to facilitate their inclusion in remote longitudinal studies include to provide participants the option to receive support to fill out surveys, which is already offered by the SMSR, invest additional communication efforts to engage these individuals in data collection and offer timely

assistance, and adapt formatting, such as text size, to enhance survey accessibility [45,46]. Nevertheless, there remains a distinct need for future studies to develop solutions that accommodate these individuals and address these disparities effectively.

Despite our study's large sample size, with available data spanning up to thirteen surveys over an eight-year follow-up period, our findings do not offer major new insights on predictors of long-term retention. The limited selection of variables used to define the clusters may have contributed to these outcomes. Nevertheless, the variables we assessed extend beyond those in similar studies on retention in pwMS, with the most comprehensive study to our knowledge being a systematic review, which found that assessments are limited to patient-related (sociodemographic data, MS type, years since MS diagnosis, disability level) and study design variables [2]. In the context of MS, considering additional contextual variables related to individuals' life and disease course could provide additional insights on long-term retention. For example, whether participants receive long-term care could offer insights into their motivation to participate or their access to support for study participation. While we did not assess this due to the small size of this group in our study, it could be valuable to explore in future research. There is also a growing body of literature highlighting the need to consider additional contextual and behavioral variables that extend beyond typical assessments of retention in chronic disease research [47–49]. For instance, a study on retention indicators across multiple remote digital health studies found that contextual factors, such as clinical referral to join a study, helped identify participant subgroups based on retention [50]. Another study found that contextualizing data of participants' mental health with the time required to respond to and complete surveys in a one year span contributed in identifying engagement subgroups [51]. An in-person study also identified behavioral and psychological predictors of loss to follow-up that go beyond sociodemographic characteristics [10]. As such, survey response patterns may be influenced by various unexplored factors. Therefore, instead of focusing solely on the analysis methods used to assess participation, attention should be directed toward developing objective metrics to predict participation based on new participant profiles, contexts, and behaviors.

In designing long-term longitudinal surveys, particularly when delivered online with limited participant-researcher interactions, it is crucial to assess factors that may increase participant burden to prevent counterproductive effects. Exploratory assessments from our study suggest a possible trend, although not statistically significant, indicating that increased survey complexity may be associated with lower survey completion. This finding is supported by the earlier mentioned systematic review of self-management interventions for pwMS, which reported a significant negative association between attrition and the length of the intervention [2]. A recent meta-analysis further highlights that design factors, such as study duration and survey length, can adversely affect participation and the usefulness of retention strategies [52]. In response, adaptive survey strategies are encouraged in the literature, where retention strategies are reviewed periodically by assessing survey non-response and collecting user feedback [53,54]. These strategies can then be modified accordingly, using engagement techniques, such as gamification or adjusting the timing and approach to data collection, to align with participants' preferences [55,56]. However, there is limited information available to apply such methodologies due to the scarcity of quantitative reports on long-term retention, which is likely a result of longitudinal studies mainly reporting missing data based on item non-response rather than long-term response and follow-up rates [57]. As remote data collection becomes more common in longitudinal studies, they continue to pose a higher risk of attrition, especially among individuals with chronic conditions who require lower-burden tasks. It is, therefore, crucial to document and apply shared experiences on methodological adaptations for improving long-term retention effectively.

## Limitations and future research

This study provides valuable and detailed insights into patterns of long-term retention in a large-scale, remote longitudinal study by exploring participant subgroups and study design elements. However, this study presents some limitations. Firstly, given the limited number of data collection points available after baseline, binary outcome variables for low and high long-term retention were derived based on approaches suggested in the literature on longitudinal studies. These derived measures may not be fully reflective of long-term retention in remote longitudinal studies. Secondly, reasons for non-response were not collected, which may limit the contextualization of our study's findings. Thirdly, specific survey design characteristics, such as the time of day when surveys were responded to, were unavailable, which limits the ability to assess possible contextual factors for survey non-response. Nevertheless, the complexity measures used in this study provide an initial context to assess possible survey design elements that may affect survey non-response.

## Conclusion

Our assessment of participants in the online arm of the Swiss Multiple Sclerosis Registry (SMSR) provides new insights into the application of well-recognized statistical methods to identify predictors of long-term retention in remote longitudinal studies. However, our findings also underscore the current difficulties in identifying participant subgroups using the limited measures generally employed in chronic disease research to assess long-term retention in studies. As our findings suggest a link between survey completion and complexity, it is crucial to identify participant subgroups and tailor survey elements to accommodate their characteristics and preferences to facilitate long-term retention. In future remote longitudinal studies, integrating additional contextual and psychological measures into surveys, and assessing them against survey response outcomes may help define these subgroups. Researchers can then adapt survey elements using various engagement strategies in response to these observations, ideally iterating with regular revisions until an optimal strategy is identified.

## Supporting information

**S1 Fig. Flowchart of included participants.**
(DOCX)

**S2 Fig. Elbow method.**
(DOCX)

**S3 Fig. Sankey plots of clustering results.**
(DOCX)

**S4 Fig. Flesch-Kinkaid score and impact on non-completion, with filtered out survey completers who answered less than the median of survey questions.**
(DOCX)

**S1 Table. Yearly retention and starting year-based retention outcomes for participants who deregistered from the SMSR or died during the study.**
(DOCX)

**S2 Table. Sensitivity analysis with alternate outcome variable, participant characteristics, yearly retention until the end of the study period.**
(DOCX)

**S3 Table. Univariate and multivariable logistic regression, yearly retention.**
(DOCX)

**S4 Table. Sensitivity analysis with therapy timing variable, univariate and multivariable logistic regression, yearly retention.**
(DOCX)

**S5 Table. Clustering task, baseline participants.**
(DOCX)

**S6 Table. Retention outcomes based on identified clusters.**
(DOCX)

**S7 Table. Univariate and multivariable logistic regression, starting year-based retention.**
(DOCX)

**S8 Table. Sensitivity analysis with alternate outcome variable, univariate and multivariable logistic regression, yearly retention until the end of the study period.**
(DOCX)

**S9 Table. Sensitivity analysis with therapy timing variable, univariate and multivariable logistic regression, starting year-based retention.**
(DOCX)

**S10 Table. Descriptions of the complexity variables for SMSR surveys.**
(DOCX)

## Author Contributions

**Conceptualization:** Paola Daniore, Viktor von Wyl.

**Data curation:** Nina Steinemann.

**Formal analysis:** Paola Daniore, Chuqiao Yan.

**Investigation:** Paola Daniore, Chuqiao Yan, Viktor von Wyl.

**Methodology:** Paola Daniore, Chuqiao Yan, Viktor von Wyl.

**Project administration:** Sabin Ammann, Nina Steinemann.

**Resources:** Nina Steinemann.

**Supervision:** Viktor von Wyl.

**Validation:** Paola Daniore, Mina Stanikic, Stefania Iaquinto, Sabin Ammann, Christian P. Kamm, Chiara Zecca, Pasquale Calabrese, Nina Steinemann, Viktor von Wyl.

**Visualization:** Paola Daniore, Chuqiao Yan.

**Writing – original draft:** Paola Daniore.

**Writing – review & editing:** Paola Daniore, Chuqiao Yan, Mina Stanikic, Stefania Iaquinto, Sabin Ammann, Christian P. Kamm, Chiara Zecca, Pasquale Calabrese, Nina Steinemann, Viktor von Wyl.

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
