## [Decision Letter · Decision Letter 0]

23 Jul 2024

PDIG-D-24-00231

Real-world patterns in remote longitudinal study participation: a study of the Swiss Multiple Sclerosis Registry

PLOS Digital Health

Dear Dr. Daniore,

Thank you for submitting your manuscript to PLOS Digital Health. After careful consideration, we feel that it has merit but does not fully meet PLOS Digital Health's publication criteria as it currently stands. Therefore, we invite you to submit a revised version of the manuscript that addresses the points raised during the review process.

Please submit your revised manuscript within 60 days Sep 21 2024 11:59PM. If you will need more time than this to complete your revisions, please reply to this message or contact the journal office at digitalhealth@plos.org. Please include the following items when submitting your revised manuscript:

We look forward to receiving your revised manuscript.

Kind regards,

Cleva Villanueva, M.D., Ph.D.

Guest Editor

PLOS Digital Health

Journal Requirements:

1. Please send a completed 'Competing Interests' statement, including any COIs declared by your co-authors. If you have no competing interests to declare, please state "The authors have declared that no competing interests exist". Otherwise please declare all competing interests beginning with the statement "I have read the journal's policy and the authors of this manuscript have the following competing interests:"

3. Please provide separate figure files in .tif or .eps format.

Additional Editor Comments (if provided):

To revise your manuscript according to the reviewer's comments and the journal guidelines of PLOS Digital Health, here is a detailed plan for each section:

Methods

 Make the Section more Compact:

 Review each paragraph and sentence to eliminate redundancy.

 Use concise language and remove any unnecessary details.

 Consider using bullet points or subheadings for clarity.

 Addressing Factors Affecting the Model:

 Underhoused Persons: Acknowledge the potential impact of underhoused individuals not completing follow-up surveys and how this might bias the results.

 Hospitalized or Institutionalized Persons: Discuss the implications of participants being in hospitals or other facilities on survey completion rates.

 Timing of SMSR Completion: Clarify whether the SMSR is completed at the time of MS diagnosis or if it includes participants diagnosed over a prolonged period. Discuss how MS treatments over time could affect the study results.

 Managing Death or De-registration Events:

 Explain how the departure of 41 patients due to death or de-registration was handled in the analysis.

 Clarify whether these patients were classified as low retention or lower response than the median and how their data were treated.

 Justification for K-means Clustering Method:

 Provide a rationale for choosing the K-means clustering method.

 Compare it with other clustering methods such as Robust and Sparse K-means or weighted K-means, or justify why these methods were not used.

 Details on Unsupervised Clustering Methods:

 Include more information about the unsupervised clustering methods and results.

 Add the elbow method graphical representation in the supplemental information.

 Validation of Clustering Analysis:

 Explain any validation methods used for the unsupervised clustering analysis.

 Discuss how the large sample size and multi-year span of the data were leveraged for validation.

Results

 Improving Figure 3:

 Make the figure more readable by increasing the size of the text.

 Consider reformatting or redesigning the figure for better clarity.

References

 Consistent Formatting:

 Ensure all references are formatted according to the journal's guidelines.

 Use a consistent style for all references, including punctuation, capitalization, and order of information.

Supporting Information

 Table of Contents:

 Create a table of contents for the supporting information section, following the journal's submission guidelines.

Reviewers' comments:

Reviewer's Responses to Questions

**Comments to the Author**

1. Does this manuscript meet PLOS Digital Health’s publication criteria? Is the manuscript technically sound, and do the data support the conclusions? The manuscript must describe methodologically and ethically rigorous research with conclusions that are appropriately drawn based on the data presented.

Reviewer #1: Yes

Reviewer #2: Yes

2. Has the statistical analysis been performed appropriately and rigorously?

Reviewer #1: Yes

Reviewer #2: No

3. Have the authors made all data underlying the findings in their manuscript fully available (please refer to the Data Availability Statement at the start of the manuscript PDF file)?

Reviewer #1: Yes

Reviewer #2: Yes

4. Is the manuscript presented in an intelligible fashion and written in standard English?

Reviewer #1: Yes

Reviewer #2: Yes

5. Review Comments to the Author

Reviewer #1: The content of the paper is quite interesting and relevant for long term retention in remote longitudinal studies. The main claims of the paper were made clear and a great amount of evidence was included in the paper. Please check the journals guidelines of submission again, for instance; in-text sources are supposed to be listed in brackets, Figures are supposed to be titled Fig, Headings and spacing needs to be changed accordingly, page numbering & line numbering is missing etc. An example sheet is provided on the official website of PLOS Digital Health. I wonder whether it is possible to reduce the method section a little bit to make it more compact. Figure 3 is hard to read, maybe it can be displayed in a better way (letters are quite small). Formatting between References 1-9 and 10 ff. is different. Supporting Information needs a table of content at the end of the manuscript (an example can also be found in the submission guidelines).

Reviewer #2: This is a study examining participant retention in a longitudinal Multiple Sclerosis study. The authors used baseline characteristics from an online survey to determine factors associated with participant retention and performed an unsupervised clustering analysis. The results of the analyses are not novel, as highlighted by the authors in the discussion. However, the results are important and add to the existing literature on this topic and does explore this topic in a longitudinal study that is of clinical importance in Switzerland. 

1) The models fail to account for several important clinical and social economic variables that are potential confounders that could impact survey completion rates. For example, underhoused persons may not receive the follow up surveys and thus be less likely to complete surveys. Persons admitted to hospital or other facility may also be less likely to complete the surveys. Is the SMSR completed at time of diagnosis of MS? Or are there participants who have had a diagnosis of MS for a prolonged period of time? If so, what are the implications of MS treatments on the study results? Does the survey data contain any information about these and other factors that may explain reasons for reduced survey completion rates? If so, this should be included in the models. If not, the authors should explain in the discussion section how these confounders may influence their results. 

2) How do the methods manage death or de-registration events in the analyses? Though only 41 patients left the SMSR by death/de-registration, the authors do not comment how these events were treated in the analyses. Were these patients all classified as low retention or lower response than median? This should be explicitly described by the authors. 

3) Why was the K-means clustering method chosen? Other clustering methods (ie. Robust and Sparse K-means, weighted K-means) may prove more advantageous in this dataset given the type of variables included. The authors should consider comparing other clustering methods in the analysis or provide a justification for the K-means methodology used. 

4) The authors provide limited information about the unsupervised clustering analysis methods and results. The elbow method graphical representation could be included in the supplemental information for example. Was there any validation of the unsupervised clustering analysis? Presumably with such a large sample size spanning multiple years, validation methods could be performed.

6. PLOS authors have the option to publish the peer review history of their article (what does this mean?). If published, this will include your full peer review and any attached files.

**Do you want your identity to be public for this peer review?** For information about this choice, including consent withdrawal, please see our Privacy Policy.

Reviewer #1: No

Reviewer #2: No

---

## [Editor Report · Decision Letter 1]

17 Sep 2024

Real-world patterns in remote longitudinal study participation: a study of the Swiss Multiple Sclerosis Registry

PDIG-D-24-00231R1

Dear Paola Daniori,

We are pleased to inform you that your manuscript 'Real-world patterns in remote longitudinal study participation: a study of the Swiss Multiple Sclerosis Registry' has been provisionally accepted for publication in PLOS Digital Health.

Best regards,

Cleva Villanueva, M.D., Ph.D.

Guest Editor

PLOS Digital Health

The manuscript is accepted for publication in PLOS Digital Health because the authors appropriately addressed all the reviewers' comments and questions and made the necessary revisions